# Bayesian Meta-Learning for the Few-Shot Setting via Deep Kernels

**Massimiliano Patacchiola**
School of Informatics
University of Edinburgh
mpatacch@ed.ac.uk

**Jack Turner**
School of Informatics
University of Edinburgh
jack.turner@ed.ac.uk

**Elliot J. Crowley**
School of Engineering
University of Edinburgh
elliot.j.crowley@ed.ac.uk

**Michael O'Boyle**
School of Informatics
University of Edinburgh
mob@inf.ed.ac.uk

**Amos Storkey**
School of Informatics
University of Edinburgh
a.storkey@ed.ac.uk

## Abstract

Recently, different machine learning methods have been introduced to tackle the challenging few-shot learning scenario that is, learning from a small labeled dataset related to a specific task. Common approaches have taken the form of meta-learning: learning to learn on the new problem given the old. Following the recognition that meta-learning is implementing learning in a multi-level model, we present a Bayesian treatment for the meta-learning inner loop through the use of deep kernels. As a result we can learn a kernel that transfers to new tasks; we call this *Deep Kernel Transfer (DKT)*. This approach has many advantages: is straightforward to implement as a single optimizer, provides uncertainty quantification, and does not require estimation of task-specific parameters. We empirically demonstrate that DKT outperforms several state-of-the-art algorithms in few-shot classification, and is the state of the art for cross-domain adaptation and regression. We conclude that complex meta-learning routines can be replaced by a simpler Bayesian model without loss of accuracy.

## 1 Introduction

One of the key differences between state-of-the-art machine learning methods, such as deep learning (LeCun et al., 2015; Schmidhuber, 2015), and human learning is that the former needs a large amount of data in order to find relevant patterns across samples, whereas the latter acquires rich structural information from a handful of examples. Moreover, deep learning methods struggle in providing a measure of uncertainty, which is a crucial requirement when dealing with scarce data, whereas humans can effectively weigh up different alternatives given limited evidence. In this regard, some authors have suggested that the human ability for few-shot inductive reasoning could derive from a Bayesian inference mechanism (Steyvers et al., 2006; Tenenbaum et al., 2011). Accordingly, we argue that the natural interpretation of meta-learning as implementing learning in a hierarchical model, leads to a Bayesian equivalent through the use of deep kernel methods.

Deep kernels combine neural networks with kernels to provide scalable and expressive closed-form covariance functions (Hinton and Salakhutdinov, 2008; Wilson et al., 2016). If one has a large number of small but related tasks, as in few-shot learning, it is possible to define a common prior that induces knowledge transfer. This prior can be a deep kernel with parameters shared across tasks, so that given a new unseen task it is possible to effectively estimate the posterior distribution over a query set conditioned on a small support set. In a meta-learning framework (Hospedales et al., 2020) this

corresponds to a Bayesian treatment for the inner loop cycle. This is our proposed approach, which we refer to as deep kernel learning with transfer, or *Deep Kernel Transfer (DKT)* for short.

We derive two versions of DKT for both the regression and the classification setting, comparing it against recent methods on a standardized benchmark environment; the code is released with an open-source license[1]. DKT has several advantages over other few-shot methods, which can be summarized as follows:

1. *Simplicity and efficiency*: it does not require any complex meta-learning optimization routines, it is straightforward to implement as a single optimizer as the inner loop is replaced by an analytic marginal likelihood computation, and it is efficient in the low-data regime.

2. *Flexibility*: it can be used in a variety of settings such as regression, cross-domain and within-domain classification, with state-of-the-art performance.

3. *Robustness*: it provides a measure of uncertainty with respect to new instances, that is crucial for a decision maker in the few-shot setting.

*Main contributions:* (i) a novel approach to deal with the few-shot learning problem through the use of deep kernels, (ii) an effective Bayesian treatment for the meta-learning inner-loop, and (iii) empirical evidence that complex meta-learning routines for few-shot learning can be replaced by a simpler hierarchical Bayesian model without loss of accuracy.

## 1.1 Motivation

The Bayesian meta-learning approach to the few-shot setting has predominantly followed the route of hierarchical modeling and multi-task learning (Finn et al., 2018; Gordon et al., 2019; Yoon et al., 2018). The underlying directed graphical model distinguishes between a set of shared parameters $\boldsymbol{\theta}$, common to all tasks, and a set of $N$ task-specific parameters $\boldsymbol{\rho}_t$. Given a train dataset of tasks $\mathcal{D} = \{\mathcal{T}_t\}_{t=1}^{N}$, each one containing input-output pairs $\mathcal{T} = \{(x_l, y_l)\}_{l=1}^{L}$, and given a test point $x_*$ from a new task $\mathcal{T}_*$, learning consists of finding an estimate of $\boldsymbol{\theta}$, forming the posterior distribution over the task-specific parameters $p(\boldsymbol{\rho}_t | x_*, \mathcal{D}, \boldsymbol{\theta})$, and then computing the posterior predictive distribution $p(y_* | x_*, \boldsymbol{\theta})$. This approach is principled from a probabilistic perspective, but is problematic, as it requires managing two levels of inference via amortized distributions or sampling, often requiring cumbersome architectures.

In recent differentiable meta-learning methods, the two sets of parameters are learned by maximum likelihood estimation, by iteratively updating $\boldsymbol{\theta}$ in an outer loop, and $\boldsymbol{\rho}_t$ in a inner loop (Finn et al., 2017). This case has various issues, since learning is destabilized by the joint optimization of two sets of parameters, and by the need to estimate higher-order derivatives (gradient of the gradient) for updating the weights (Antoniou et al., 2019).

To avoid these drawbacks we propose a simpler solution, that is marginalizing $\boldsymbol{\rho}_t$ over the data of a specific task. This marginalization is analytic and leads to a closed form marginal likelihood, which measures the *expectedness* of the data under the given set of parameters. By finding the parameters of a deep kernel we can maximize the marginal likelihood. Following our approach there is no need to estimate the posterior distribution over the task-specific parameters, meaning that it is possible to directly compute the posterior predictive distribution, skipping an intermediate inference step. We argue that this approach can be very effective in the few-shot setting, significantly reducing the complexity of the model with respect to meta-learning approaches, while retaining the advantages of Bayesian methods (e.g. uncertainty estimation) with state-of-the-art performances.

## 2 Background

### 2.1 Few-shot Learning

The terminology describing the few-shot learning setup is dispersive due to the colliding definitions used in the literature; the reader is invited to see Chen et al. (2019) for a comparison. Here, we use the nomenclature derived from the meta-learning literature which is the most prevalent at time of writing. Let $\mathcal{S} = \{(x_l, y_l)\}_{l=1}^{L}$ be a *support-set* containing input-output pairs, with $L$ equal to one (1-shot)

or five (5-shot), and $\mathcal{Q} = \{(x_m, y_m)\}_{m=1}^M$ be a *query-set* (sometimes referred to in the literature as a target-set), with $M$ typically one order of magnitude greater than $L$. For ease of notation the support and query sets are grouped in a *task* $\mathcal{T} = \{\mathcal{S}, \mathcal{Q}\}$, with the dataset $\mathcal{D} = \{\mathcal{T}_t\}_{t=1}^N$ defined as a collection of such tasks. Models are trained on random tasks sampled from $\mathcal{D}$, then given a new task $\mathcal{T}_* = \{\mathcal{S}_*, \mathcal{Q}_*\}$ sampled from a test set, the objective is to condition the model on the samples of the support $\mathcal{S}_*$ to estimate the membership of the samples in the query set $\mathcal{Q}_*$. In the most common scenario, training, validation and test datasets each consist of distinct tasks sampled from the same overall distribution over tasks. Note that the target value $y$ can be a continuous value (regression) or a discrete one (classification), though most previous work has focused on classification. We also consider the *cross-domain* scenario, where the test tasks are sampled from a different distribution over tasks than the training tasks; this is potentially more representative of many real-world scenarios.

## 2.2 Kernels

Given two input instances $x$ and $x'$ and a function $f(\cdot)$, the kernel $k(x, x')$ is a covariance function that expresses how the correlation of the outputs at two points depends on the relationship between their two locations in input space

$$k(x, x') = \text{cov}(f(x), f(x')). \tag{1}$$

The simplest kernel has a linear expression $k_{\text{LIN}}(x, x') = v\langle x, x'\rangle$, where $\langle \cdot \rangle$ denotes an inner product, and $v$ is a variance hyperparameter. The use of a linear kernel is computationally convenient and it induces a form of Bayesian linear regression, however this is often too simplistic. For this reason, a variety of other kernels has been proposed in the literature: Radial Basis Function kernel (RBF), Matérn kernel, Cosine Similarity kernel (CosSim), and the spectral mixture kernel (Wilson and Adams, 2013). Details about the kernels used in this work are given in Appendix A (supp. material).

In deep kernel learning (Hinton and Salakhutdinov, 2008; Wilson et al., 2016) an input vector $\mathbf{x}$ is mapped to a latent vector $\mathbf{h}$ through a non-linear function $\mathcal{F}_\phi(\mathbf{x}) \rightarrow \mathbf{h}$ (e.g. a neural network) parameterized by a set of weights $\phi$. The embedding is defined such that the dimensionality of the input is significantly reduced, meaning that if $\mathbf{x} \in \mathbb{R}^J$ and $\mathbf{h} \in \mathbb{R}^K$ then $J \gg K$. Once the input has been encoded in $\mathbf{h}$ the latent vector is passed to a kernel. When the inputs are images a common choice for $\mathcal{F}_\phi$ is a Convolutional Neural Network (CNN). Specifically we construct a kernel

$$k(\mathbf{x}, \mathbf{x}'|\boldsymbol{\theta}, \boldsymbol{\phi}) = k'(\mathcal{F}_\phi(\mathbf{x}), \mathcal{F}_\phi(\mathbf{x}')|\boldsymbol{\theta}) \tag{2}$$

from some latent space kernel $k'$ with hyperparameters $\boldsymbol{\theta}$ by passing the inputs through the non-linear function $\mathcal{F}_\phi$. Then the hyperparameters $\boldsymbol{\theta}$ and the parameters of the model $\phi$ are jointly learned by maximizing the log marginal likelihood, backpropagating the error.

## 3 Description of the method

Let us start from the interpretation of meta-learning as a hierarchical model (Finn et al., 2018; Grant et al., 2018), considering a set of task-common parameters in the upper hierarchy (optimized in an outer loop), along with a process for determining task-specific parameters in the lower hierarchy (optimized in an inner loop). For example, in MAML (Finn et al., 2017), the outer-parameters are the common neural network initialization weights and the inner-parameters are the final network weights, with prior implicitly defined by the probability that a particular parameterization can be reached in a few gradient steps from the initial parameters. Both outer and inner loops are obtained end-to-end, by differentiating through the inner loop to obtain derivatives for the outer loop parameters. This causes well known instability problems (Antoniou et al., 2019).

**Algorithm 1** Deep Kernel Transfer (DKT) in the few-shot setting, train and test functions.
***
**Require:** $\mathcal{D} = \{\mathcal{T}_n\}_{n=1}^N$ train dataset and $\mathcal{T}_* = \{\mathcal{S}_*, \mathcal{Q}_*\}$ test task.
**Require:** $\hat{\boldsymbol{\theta}}$ kernel hyperparameters and $\hat{\boldsymbol{\phi}}$ neural network weights. &emsp;&emsp;&emsp;&emsp; ▷ Randomly initialized
**Require:** $\alpha, \beta$: step size hyperparameters for the optimizers.
***
1: **function** TRAIN($\mathcal{D}, \alpha, \beta, \hat{\boldsymbol{\theta}}, \hat{\boldsymbol{\phi}}$)
2: &emsp;&emsp; **while** not done **do**
3: &emsp;&emsp;&emsp; Sample task $\mathcal{T} = \{\mathcal{S}, \mathcal{Q}\} \sim \mathcal{D}$
4: &emsp;&emsp;&emsp; Assign $\mathcal{T}^x \leftarrow \forall x \in \mathcal{S} \cup \mathcal{Q}$ and $\mathcal{T}^y \leftarrow \forall y \in \mathcal{S} \cup \mathcal{Q}$
5: &emsp;&emsp;&emsp; $\mathcal{L} = -\log p(\mathcal{T}^y | \mathcal{T}^x, \hat{\boldsymbol{\theta}}, \hat{\boldsymbol{\phi}})$ &emsp;&emsp;&emsp;&emsp;&emsp;&emsp;&emsp;&emsp;&emsp;&emsp; ▷ see Eq. (3)
6: &emsp;&emsp;&emsp; Update $\hat{\boldsymbol{\theta}} \leftarrow \hat{\boldsymbol{\theta}} - \alpha\nabla_{\hat{\theta}}\mathcal{L}$ and $\hat{\boldsymbol{\phi}} \leftarrow \hat{\boldsymbol{\phi}} - \beta\nabla_{\hat{\phi}}\mathcal{L}$
7: &emsp;&emsp; **end while**
8: &emsp;&emsp; **return** $\hat{\boldsymbol{\theta}}, \hat{\boldsymbol{\phi}}$
9: **end function**
10: **function** TEST($\mathcal{T}_*, \hat{\boldsymbol{\theta}}, \hat{\boldsymbol{\phi}}$)
11: &emsp;&emsp; Assign $\mathcal{T}_*^x \leftarrow \forall x \in \mathcal{S}_*, \mathcal{T}_*^y \leftarrow \forall y \in \mathcal{S}_*$, and $x_* \leftarrow x \in \mathcal{Q}_*$
12: &emsp;&emsp; **return** $p(y_* | x_*, \mathcal{T}_*^x, \mathcal{T}_*^y, \hat{\boldsymbol{\theta}}, \hat{\boldsymbol{\phi}})$ &emsp;&emsp;&emsp;&emsp;&emsp;&emsp;&emsp;&emsp; ▷ see Eq. (5)
13: **end function**
***

Our proposal is to replace the inner loop with a Bayesian integral, while still optimizing for the parameters. This is commonly called a *maximum likelihood type II (ML-II)* approach. Specifically, we learn a set of parameters and hyperparameters of a deep kernel (outer-loop) that maximize a marginal likelihood across all tasks. The marginalization of this likelihood integrates out over each of the task-specific parameters for each task using a Gaussian process approach, replacing the inner loop model with a kernel.

Let all the input data (support and query) for task $t$ be denoted by $\mathcal{T}_t^x$ and the target data be $\mathcal{T}_t^y$. Let $\mathcal{D}^x$ and $\mathcal{D}^y$ denote the respective collections of these datasets over all tasks; this data is hierarchically grouped by task. The marginal-likelihood of a Bayesian hierarchical model, conditioned on task-common hyperparameters $\hat{\boldsymbol{\theta}}$ and other task-common parameters $\hat{\boldsymbol{\phi}}$ (e.g. neural network weights) would take the form

$$P(\mathcal{D}^y | \mathcal{D}^x, \hat{\boldsymbol{\theta}}, \hat{\boldsymbol{\phi}}) = \prod_t P(\mathcal{T}_t^y | \mathcal{T}_t^x, \hat{\boldsymbol{\theta}}, \hat{\boldsymbol{\phi}}), \tag{3}$$

where $P(\mathcal{T}_t^y | \mathcal{T}_t^x, \hat{\boldsymbol{\theta}}, \hat{\boldsymbol{\phi}})$ is a marginalization over each set of task-specific parameters. Let these task-specific parameters for task $t$ be denoted by $\boldsymbol{\rho}_t$, then

$$P(\mathcal{T}_t^y | \mathcal{T}_t^x, \boldsymbol{\theta}, \boldsymbol{\phi}) = \int \prod_k P(y_k | x_k, \boldsymbol{\theta}, \boldsymbol{\phi}, \boldsymbol{\rho}_t) d\boldsymbol{\rho}_t, \tag{4}$$

where $k$ enumerates elements of $x_k \in \mathcal{T}_t^x$, and corresponding elements $y_k \in \mathcal{T}_t^y$. In typical meta-learning, the task-specific integral (4) would be replaced by an inner-loop optimizer for the task-specific objective (and the parameters of that optimizer); any additional cross-task parameters $\boldsymbol{\theta}$, $\boldsymbol{\phi}$ would be optimized in the outer loop. Instead, we do a full integral of the task specific parameters, and optimize only for the cross-task parameters $\boldsymbol{\theta}, \boldsymbol{\phi}$. We do that implicitly rather than explicitly by using a Gaussian process model for $P(\mathcal{T}_t^y | \mathcal{T}_t^x, \boldsymbol{\theta})$, which is the outcome of an analytic integral of Equation 4 for many model classes (Rasmussen and Williams, 2006). Predictions of the value $y_*$ for a new point $x_*$ given a small set of exemplars $\mathcal{T}_{t_*}^x, \mathcal{T}_{t_*}^y$ for a new task $t_*$ can be made using the predictive distribution

$$p(y_* | x_*, \mathcal{T}_{t_*}^x, \mathcal{T}_{t_*}^y) \approx p(y_* | x_*, \mathcal{T}_{t_*}^x, \mathcal{T}_{t_*}^y, \hat{\boldsymbol{\theta}}, \hat{\boldsymbol{\phi}}). \tag{5}$$

Our claim is that, though the number of data points for each task is potentially small, the total number of points over all tasks contributing to the marginal likelihood (3) is sufficiently large to make ML-II appropriate for finding a set of shared weights and parameters without underfitting or overfitting. Those parameters provide a model with good generalization capability to new unseen tasks, removing the need for inferring the task-specific parameters $\boldsymbol{\rho}_t$. Results in Section 5 indicate that our proposal is competitive with much more complicated meta-learning methods. Note that this approach differs from direct deep kernel learning, where the marginalization is over all data; this would ignore the task distinctions which is vital given the hierarchical mode (see experimental comparison in Section 5.1). The problem also differs from multitask learning where the tasks share the same input values.

For stochastic gradient training, at each iteration, a task $\mathcal{T} = \{\mathcal{S}, \mathcal{Q}\}$ is sampled from $\mathcal{D}$, then the log marginal likelihood, that is the logarithm of (3), is estimated over $\mathcal{S} \cup \mathcal{Q}$ (assuming $y \in Q$ to be observed) and the parameters of the kernel are updated via a gradient step on the marginal likelihood objective for that task. This procedure allows us to find a kernel that can represent the task in its entirety over both support and query sets. At test time, given a new task $\mathcal{T}_* = \{\mathcal{S}_*, \mathcal{Q}_*\}$ the prediction on the query set $\mathcal{Q}_*$ is made via conditioning on the support set $\mathcal{S}_*$, using the parameters that have been learned at training time. Pseudocode is given in Algorithm 1.

## 3.1 Regression

We want to find a closed form expression of (3) for the regression case. Assume we are interested in a continuous output $y_*$ generated by a clean signal $f_*(x_*)$ corrupted by homoscedastic Gaussian noise $\epsilon$ with variance $\sigma^2$. We are interested in the joint distribution of the observed outputs and the function values at test location. For ease of notation, let us define $\mathbf{k}_* = k(x_*, \mathbf{x})$ to denote the $N$-dimensional vector of covariances between $x_*$ and the $N$ training points in $\mathcal{D}$. Similarly, let us write $k_{**} = k(x_*, x_*)$ for the variance of $x_*$, and $\mathbf{K}$ to identify the covariance matrix on the training inputs in $\mathcal{D}$. The predictive distribution $p(y_* | x_*, \mathcal{D})$ is obtained by Bayes' rule, and given the conjugacy of the prior, this is a Gaussian with mean and covariance specified as

$$\mathbb{E}[f_*] = \mathbf{k}_*^\top (\mathbf{K} + \sigma^2 \mathbf{I})^{-1} \mathbf{y}, \tag{6a}$$

$$\text{cov}(f_*) = k_{**} - \mathbf{k}_*^\top (\mathbf{K} + \sigma^2 \mathbf{I})^{-1} \mathbf{k}_*. \tag{6b}$$

Note that (6) defines a distribution over functions, which assumes that the collected values at any finite set of points have a joint Gaussian distribution (Rasmussen and Williams, 2006). Hereon, we absorb the noise $\sigma^2 \mathbf{I}$ into the covariance matrix $\mathbf{K}$ and treat it as part of a vector of learnable parameters $\boldsymbol{\theta}$, that also include the hyperparameters of the kernel (e.g. variance of a linear kernel).

Let us collect all the target data items for task $t$ into vector $\mathbf{y}_t$, and denote the kernel between all task inputs by $K_t$. It follows that the marginal likelihood of Equation (3) can be rewritten as

$$\log P(\mathcal{D}^y | \mathcal{D}^x, \hat{\boldsymbol{\theta}}, \hat{\boldsymbol{\phi}}) = \sum_t \underbrace{-\frac{1}{2} \mathbf{y}_t^\top [K_t(\hat{\boldsymbol{\theta}}, \hat{\boldsymbol{\phi}})]^{-1} \mathbf{y}_t}_{\text{data-fit}} \underbrace{- \frac{1}{2} \log |K_t(\hat{\boldsymbol{\theta}}, \hat{\boldsymbol{\phi}})|}_{\text{penalty}} + c, \tag{7}$$

where $c$ is a constant. The parameters are estimated via ML-II maximizing (7) via gradient ascent. In practice we use a stochastic gradient ascent with each batch containing the data for a single task.

## 3.2 Classification

A Bayesian treatment for the classification case does not come without problems, since a non-Gaussian likelihood breaks the conjugacy. For instance, in the case of binary classification the Bernoulli likelihood induces an intractable marginalization of the evidence and therefore it is not possible to estimate the posterior in a closed form. Common approaches to deal with this issue (e.g. MCMC or variational methods), incur a significant computational cost for few-shot learning: for each new task, the posterior is estimated by approximation or sampling, introducing an inner loop that increases the time complexity from constant $\mathcal{O}(1)$ to linear $\mathcal{O}(K)$, with $K$ being the number of inner cycles. An alternative solution would be to treat the classification problem as if it were a regression one, therefore reverting to analytical expressions for both the evidence and the posterior. In the literature this has been called *label regression (LR)* (Kuss, 2006) or *least-squares classification (LSC)* (Rifkin and Klautau, 2004; Rasmussen and Williams, 2006). Experimentally, LR and LSC tend to be more effective than other approaches in both binary (Kuss, 2006) and multi-class (Rifkin and Klautau, 2004) settings. Here, we derive a classifier based on LR which is computationally cheap and straightforward to implement.

Let us define a binary classification setting, with the class being a Bernoulli random variable $c \in \{0, 1\}$. The model is trained as a regressor with a target $y_+ = 1$ to denote the case $c = 1$, and $y_- = -1$ to denote the case $c = 0$. Even though $y \in \{-1, 1\}$ there is no guarantee that $f(x) \in [y_-, y_+]$. Predictions are made by computing the predictive mean and passing it through a sigmoid function, inducing a probabilistic interpretation. Note that it is still possible to use ML-II to make point estimates of $\boldsymbol{\theta}$ and $\phi$. When generalizing to a multi-label task we apply the *one-versus-rest* scheme where $C$ binary classifiers are used to classify each class against all the rest. The log marginal

likelihood, that is the logarithm of Equation (3), is replaced by the sum of the marginals for each one of the $C$ individual class outputs $\mathbf{y}_c$, as

$$\log p(\mathbf{y}|\mathbf{x}, \hat{\boldsymbol{\theta}}, \hat{\boldsymbol{\phi}}) = \sum_{c=1}^{C} \log p(\mathbf{y}_c|\mathbf{x}, \hat{\boldsymbol{\theta}}, \hat{\boldsymbol{\phi}}). \tag{8}$$

Given a new input $x_*$ and the $C$ outputs of all the binary classifiers, a decision is made by selecting the output with the highest probability $c_* = \mathrm{argmax}_c\big(\sigma(m_c(x_*))\big)$, where $m(x)$ is the predictive mean, $\sigma(\cdot)$ the sigmoid function, and $c_* \in \{1, ..., C\}$.

## 4 Related Work

There is a wealth of literature on feature transfer (Pan and Yang, 2009). As a baseline for few-shot learning, the standard procedure consists of two phases: pre-training and fine-tuning. During pre-training, a network and classifier are trained on examples for the base classes. When fine-tuning, the network parameters are fixed and a new classifier is trained on the novel classes. This approach has its limitations; part of the model has to be trained from scratch for each new task, and often overfits. Chen et al. (2019) extend this by proposing the use of cosine distance between examples (called Baseline++). However, this still relies on the assumption that a fixed fine-tuning protocol will balance the bias-variance tradeoff correctly for every task.

Alternatively, one can compare new examples in a learned metric space. Matching Networks (MatchingNets, Vinyals et al., 2016) use a softmax over cosine distances as an attention mechanism, and an LSTM to encode the input in the context of the support set, considered as a sequence. Prototypical Networks (ProtoNets, Snell et al., 2017) are based on learning a metric space in which classification is performed by computing distances to prototypes, where each prototype is the mean vector of the embedded support points belonging to its class. Relation Networks (RelationNets, Sung et al., 2018) use an embedding module to generate representations of the query images that are compared by a relation module with the support set, to identify matching categories.

Meta-learning (Bengio et al., 1992; Schmidhuber, 1992; Hospedales et al., 2020) methods have become very popular for few-shot learning tasks. MAML (Finn et al., 2017) has been proposed as a way to meta-learn the parameters of a model over many tasks, so that the initial parameters are a good starting point from which to adapt to a new task. MAML has provided inspiration for numerous meta-learning approaches (Antoniou et al., 2019; Rajeswaran et al., 2019).

In several works, MAML has been interpreted as a Bayesian hierarchical model (Finn et al., 2018; Grant et al., 2018; Jerfel et al., 2019). Bayesian MAML (Yoon et al., 2018) combines efficient gradient-based meta-learning with nonparametric variational inference, while keeping an application-agnostic approach. Gordon et al. (2019) have recently proposed an amortization network—VERSA—that takes few-shot learning datasets as inputs, and outputs a distribution over task-specific parameters which can be used to meta-learn probabilistic inference for prediction. Xu et al. (2019) have used conditional neural processes with an encoder-decoder architecture to project labeled data into an infinite-dimensional functional representation.

For the regression case Harrison et al. (2018) have proposed a method named ALPaCA, which uses a dataset of sample functions to learn a domain-specific encoding and a prior over weights. Tossou et al. (2019) have presented a variant of kernel learning for Gaussian Processes called Adaptive Deep Kernel Learning (ADKL), which finds a kernel for each task using a task encoder network. The difference between our method and ADKL is that we do not need an additional module for task encoding as we can rely on a single set of shared general-purpose hyperparameters.

## 5 Experiments

In the few-shot setting a fair comparison between methods is often obfuscated by substantial differences in the implementation details of each algorithm. Chen et al. (2019) have recently released an open-source benchmark to allow for a fair comparison between methods. We integrated our algorithm into this framework using PyTorch and GPyTorch (Gardner et al., 2018). In all experiments the proposed method is marked as *DKT*. Training details are reported in Appendix B (supp. material).

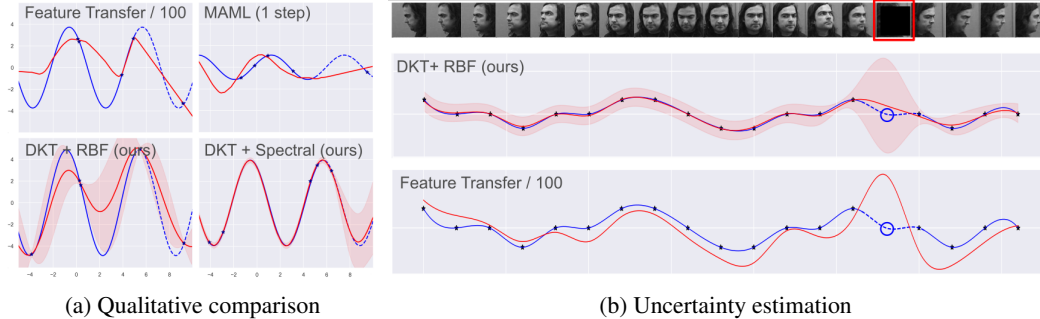

|(a) Qualitative comparison|(b) Uncertainty estimation|

Figure 1: (a) Comparison between different methods for unknown function approximation (out-of-range, 5 support points). DKT better fits (red line) the true function (solid blue) and the out-of-bound portion never seen at training time (dashed blue). Uncertainty (red shadow) increases in low-confidence regions. (b) Uncertainty estimation for an outlier (Cutout noise, red frame) in the head trajectory estimation from images. DKT is able to estimate a mean value (red line) close to the true value (blue circle) showing large variance. Feature transfer performs poorly at the same location.

## 5.1 Regression

We consider two tasks: amplitude prediction for unknown periodic functions, and head pose trajectory estimation from images. The former was treated as a few-shot regression problem by Finn et al. (2017) to motivate MAML: support and query scalars are uniformly sampled from a periodic wave with amplitude $\in [0.1, 5.0]$, phase $\in [0, \pi]$, and range $\in [-5.0, 5.0]$, and Gaussian noise ($\mu = 0$, $\sigma = 0.1$). The training set is composed of 5 support and 5 query points, and the test set of 5 support and 200 query points. We first test *in-range*: the same domain as the training set as in Finn et al. (2017). We also consider a more challenging *out-of-range* regression, with test points drawn from an extended domain $[-5.0, 10.0]$ where portions from the range $[5.0, 10.0]$ have *not been seen* at training time.

For head pose regression, we used the Queen Mary University of London multiview face dataset (QMUL, Gong et al., 1996), it comprises of grayscale face images of 37 people (32 train, 5 test). There are 133 facial images per person, covering a viewsphere of $\pm 90°$ in yaw and $\pm 30°$ in tilt at $10°$ increment. Each task consists of randomly sampled trajectories taken from this discrete manifold, where *in-range*

Table 1: Average Mean-Squared Error (MSE) and standard deviation (three runs) on few-shot regression for periodic functions (top) and head pose trajectory (bottom), using 10 samples for train, and 5 for test. Same domain is marked as *in-range*, extended unseen domain as *out-of-range*. Lowest error in bold. *Results reported by Tossou et al. (2019).

| Method | in-range | out-of-range |
|---|---|---|
| *Periodic functions* | | |
| **ADKL** (Tossou et al., 2019)* | 0.14 | – |
| **R2-D2** (Bertinetto et al., 2019)* | 0.46 | – |
| **ALPaCA** (Harrison et al., 2018) | $0.14 \pm 0.09$ | $5.92 \pm 0.11$ |
| **Feature Transfer/1** | $2.94 \pm 0.16$ | $6.13 \pm 0.76$ |
| **Feature Transfer/100** | $2.67 \pm 0.15$ | $6.94 \pm 0.97$ |
| **MAML (1 step)** | $2.76 \pm 0.06$ | $8.45 \pm 0.25$ |
| **DKBaseline + RBF** | $2.85 \pm 1.14$ | $3.65 \pm 1.63$ |
| **DKBaseline + Spectral** | $2.08 \pm 2.31$ | $4.11 \pm 1.92$ |
| **DKT + RBF** (ours) | $1.38 \pm 0.03$ | $2.61 \pm 0.16$ |
| **DKT + Spectral** (ours) | $\mathbf{0.08 \pm 0.06}$ | $\mathbf{0.10 \pm 0.06}$ |
| *Head pose trajectory* | | |
| **Feature Transfer/1** | $0.25 \pm 0.04$ | $0.20 \pm 0.01$ |
| **Feature Transfer/100** | $0.22 \pm 0.03$ | $0.18 \pm 0.01$ |
| **MAML (1 step)** | $0.21 \pm 0.01$ | $0.18 \pm 0.02$ |
| **DKT + RBF** (ours) | $0.12 \pm 0.04$ | $0.14 \pm 0.03$ |
| **DKT + Spectral** (ours) | $\mathbf{0.10 \pm 0.01}$ | $\mathbf{0.11 \pm 0.02}$ |

includes the full manifold and *out-of-range* allows training only on the leftmost 10 angles, and testing on the full manifold; the goal is to predict tilt. For the periodic function prediction experiment, we compare our approach against feature transfer and MAML (Finn et al., 2017). Moreover we report the results of ADKL (Tossou et al., 2019), R2-D2 (Bertinetto et al., 2019), and ALPaCA (Harrison et al., 2018) obtained on a similar task (as defined in Yoon et al., 2018). To highlight the importance of kernel transfer, we add a baseline where a deep kernel is trained from scratch on the support points of every incoming task without transfer (DKBaseline), this correspond to standard deep kernel learning (Wilson et al., 2016). Few methods have tackled few-shot regression from images, so in the head pose trajectory estimation we compare against feature transfer and MAML. As metric we use the average Mean-Squared Error (MSE) between predictions and true values. Additional details are reported in Appendix B.

Results for the regression experiments are summarized in Table 1 and a qualitative comparison is provided in Figure 1a and in the supplementary material. DKT obtains a lower MSE than feature

transfer and MAML on both experiments. For unknown periodic function estimation, using a spectral kernel gives a large advantage over RBF, being more precise in both in-range and out-of-range (1.38 vs 0.08, and 2.61 vs 0.10 MSE). Uncertainty is correctly estimated in regions with low point density, and increases overall in the out-of-range region. Conversely, feature transfer severely underfits (1 step, 2.94 MSE) or overfits (100 step, 2.67), and was unable to model out-of-range points (6.13 and 6.94). MAML is effective in-range (2.76), but significantly worse out-of-range (8.45). ADKL, R2-D2, and ALPaCA (0.14, 0.46, 0.14) are better than DKT with an RBF kernel (1.38), but worse than DKT with a Spectral kernel (0.08). This indicates that the combination of an appropriate kernel with our method is more effective than an adaptive approach. The DKBaseline performs significantly worse than DKT in all conditions, confirming the necessity of using kernel transfer for few-shot problems. Qualitative comparison in Figure 1a shows that both feature transfer and MAML are unable to fit the true function, especially out-of-range; additional samples are reported in Appendix C. We observe similar results for head pose estimation, with DKT reporting lower MSE in all cases (Table 1). In Appendix C we also examine the latent spaces generated by the RBF and spectral kernel.

**Uncertainty quantification (regression)** In the low-data regime it is fundamental to account for uncertainty in the prediction; DKT is one of the few methods able to do it. To highlight the benefits of our method versus other approaches, we perform an experiment on *quantifying uncertainty*, sampling head pose trajectories and corrupting one input with Cutout (DeVries and Taylor, 2017), randomly covering $95\%$ of the image. Qualitative results are shown in Figure 1b. For the corrupted input, DKT predicts a value close to the true one, while giving a high level of uncertainty (red shadow). Feature transfer performs poorly, predicting an unrealistic pose.

## 5.2 Classification

We consider two challenging datasets: the Caltech-UCSD Birds (CUB-200, Wah et al., 2011), and mini-ImageNet (Ravi and Larochelle, 2017). All the experiments are 5-way (5 random classes) with 1 or 5-shot (1 or 5 samples per class in the support set). A total of 16 samples per class are provided for the query set. Additional details in Appendix B (supp. material). We compare the following kernels: linear, RBF, Matérn, Polynomial, CosSim, and BNCosSim. Where BNCosSim is a variant of CosSim with features centered through BatchNorm (BN) statistics (Ioffe and Szegedy, 2015), this has shown to improve performance (Wang et al., 2019). We compare our approach to several state-of-the-art methods, such as MAML (Finn et al., 2017), ProtoNets (Snell et al., 2017), MatchingNet (Vinyals et al., 2016), and RelationNet (Sung et al., 2018). We further compare against feature transfer, and Baseline++ from Chen et al. (2019). All these methods have been trained from scratch with the same backbone and learning schedule. We additionally report the results for approaches with comparable training procedures and convolutional architectures (Mishra et al., 2018; Ravi and Larochelle, 2017; Wang et al., 2019) including recent hierarchical Bayesian methods (Gordon et al., 2019; Grant et al., 2018; Jerfel et al., 2019). We have excluded approaches that use deeper backbones or more sophisticated learning schedules (Antoniou and Storkey, 2019; Oreshkin et al., 2018; Qiao et al., 2018; Ye et al., 2018) so that the quality of the algorithms can be assessed separately from the power of the underlying discriminative model.

We report the results for the more challenging 1-shot case in Table 2 and 3, and the results for the 5-shot case in the supplementary material. DKT achieves the highest accuracy in both CUB (63.37%) and mini-ImageNet (49.73%), performing better than any other approach including hierarchical Bayesian methods such as LLAMA (49.40%) and VERSA (48.53%). The best performance of first-order kernels (Table 5, supp. material) is likely due to a low-curvature manifold induced by the neural network in the latent space, increasing the linear separability of data. Overall our results confirm the findings of Chen et al. (2019) regarding the effectiveness of cosine metrics, and those of Wang et al. (2019) on the importance of feature normalization (Appendix D and E). In Table 6 (supp. material) we report results with a deeper backbone (ResNet-10, He et al. 2016), showing that DKT outperforms all other methods in 5-shot (85.64%) with the second best result in 1-shot (72.27%). The difference in performance between CosSim and BNCosSim is larger for the deeper backbone, indicating that centering the features is important when additional layers are added to the network.

**Uncertainty quantification (classification)** We provide results on model calibration on the CUB dataset. We followed the protocol of Guo et al. (2017) estimating the Expected Calibration Error (ECE), a scalar summary statistic (the lower the better). We first scaled each model output, calibrating the temperature by minimizing the NLL on logits/labels via LBFGS on 3000 tasks; then we estimated

Table 2: Average accuracy and standard deviation (percentage) over three runs (1-shot, 5-ways, Conv-4) on CUB and cross-domain classification (Omniglot to EMNIST and mini-Imagenet to CUB). Best results highlighted in bold.

| Method | CUB | Omni→EMNIST | ImgNet→CUB |
|---|---|---|---|
| Feature Transfer | $46.19 \pm 0.64$ | $64.22 \pm 1.24$ | $32.77 \pm 0.35$ |
| **Baseline++** (Chen et al., 2019) | $61.75 \pm 0.95$ | $56.84 \pm 0.91$ | $39.19 \pm 0.12$ |
| **MatchingNet** (Vinyals et al., 2016) | $60.19 \pm 1.02$ | $75.01 \pm 2.09$ | $36.98 \pm 0.06$ |
| **ProtoNet** (Snell et al., 2017) | $52.52 \pm 1.90$ | $72.04 \pm 0.82$ | $33.27 \pm 1.09$ |
| **MAML** (Finn et al., 2017) | $56.11 \pm 0.69$ | $72.68 \pm 1.85$ | $34.01 \pm 1.25$ |
| **RelationNet** (Sung et al., 2018) | $62.52 \pm 0.34$ | $75.62 \pm 1.00$ | $37.13 \pm 0.20$ |
| **DKT + Linear** (ours) | $60.23 \pm 0.76$ | $\mathbf{75.97 \pm 0.70}$ | $38.72 \pm 0.42$ |
| **DKT + CosSim** (ours) | $\mathbf{63.37 \pm 0.19}$ | $73.06 \pm 2.36$ | $\mathbf{40.22 \pm 0.54}$ |
| **DKT + BNCosSim** (ours) | $62.96 \pm 0.62$ | $75.40 \pm 1.10$ | $40.14 \pm 0.18$ |

Table 3: Classification: 1-shot, 5-ways, Conv-4. Best results in bold.

| Method | mini-ImageNet |
|---|---|
| **ML-LSTM** (Ravi and Larochelle, 2017) | $43.44 \pm 0.77$ |
| **SNAIL** (Mishra et al., 2018) | $45.10$ |
| **iMAML-HF** (Rajeswaran et al., 2019) | $49.30 \pm 1.88$ |
| **LLAMA** (Grant et al., 2018) | $49.40 \pm 1.83$ |
| **VERSA** (Gordon et al., 2019) | $48.53 \pm 1.84$ |
| **Amortized VI** (Gordon et al., 2019) | $44.13 \pm 1.78$ |
| **Meta-Mixture** (Jerfel et al., 2019) | $49.60 \pm 1.50$ |
| **SimpleShot** (Wang et al., 2019) | $49.69 \pm 0.19$ |
| **Feature Transfer** | $39.51 \pm 0.23$ |
| **Baseline++** (Chen et al., 2019) | $47.15 \pm 0.49$ |
| **MatchingNet** (Vinyals et al., 2016) | $48.25 \pm 0.65$ |
| **ProtoNet** (Snell et al., 2017) | $44.19 \pm 1.30$ |
| **MAML** (Finn et al., 2017) | $45.39 \pm 0.49$ |
| **RelationNet** (Sung et al., 2018) | $48.76 \pm 0.17$ |
| **DKT + CosSim** (ours) | $48.64 \pm 0.45$ |
| **DKT + BNCosSim** (ours) | $\mathbf{49.73 \pm 0.07}$ |

the ECE on the test set. The complete results for CUB 1-shot and 5-shot (percentage, average of three runs) are reported in Appendix D, Table 7. In 1-shot DKT achieves one of the lowest ECE $2.6\%$ beating most of the competitors (only ProtoNet and MAML do better). In 5-shot our model achieves the second lowest ECE $1.1\%$ (ProtoNet does marginally better).

## 5.3 Cross-domain classification

The objective of cross-domain classification is to train a model on tasks sampled from one distribution, that then generalizes to tasks sampled from a different distribution. Specifically, we combine datasets so that the training split is drawn from one, and the validation and test split are taken from another. We experiment on mini-ImageNet→CUB (train split from mini-ImageNet and val/test split from CUB) and Omniglot→EMNIST. We compare our method to the previously-considered approaches, using identical settings for number of epochs and model selection strategy (see Appendix B). Results for the 1-shot case are given in Table 2. DKT achieves the highest accuracy in most conditions. In Omniglot→EMNIST, the best performance is achieved with a linear kernel (75.97%). In mini-ImageNet→CUB, DKT surpasses all the other methods obtaining the highest accuracy with CosSim (40.22%) and BNCosSim (40.14%). Note that most competing methods experience difficulties in this setting, as shown by their low accuracies and large standard deviations. A comparison of kernels shows that first order ones are more effective (see Appendix E, Table 9).

## 6 Conclusion

In this work, we have introduced DKT, a highly flexible Bayesian model based on deep kernel learning. Compared with some other approaches in the literature, DKT performs better in regression and cross-domain classification while providing a measure of uncertainty. Based on the results, we argue that many complex meta-learning routines for few-shot learning can be replaced by a simple hierarchical Bayesian model without loss of accuracy. Future work could focus on exploiting the flexibility of the model in related settings, especially those merging continual and few-shot learning (Antoniou et al., 2020), where DKT has the potential to thrive.

## Broader Impact

The main motivation of this work has been to design a simple yet effective Bayesian method for dealing with the few-shot learning setting. The ability to learn from a reduced amount of data is crucial if we want to have systems that are able to deal with concrete real-world problems. Applications include (but are not limited to): classification and regression under constrained computational resources, medical diagnosis from small datasets, biometric identification from a handful of images, etc. Our method is one of the few which is able to provide a measure of uncertainty as a feedback for the decision maker. However, it is important to wisely choose the data on which the system is trained, since the low-data regime may be prone to bias more than the standard counterpart. If data is biased our method is not guaranteed to provide a correct estimation; this could harm the final users and should be carefully taken into account.

## Acknowledgments and Disclosure of Funding

This work was supported by a Huawei DDMPLab Innovation Research Grant. This project has received funding from the European Union's Horizon 2020 research and innovation programme under grant agreement No 732204 (Bonseyes).This work is supported by the Swiss State Secretariat for Education, Research and Innovation (SERI) under contract number 16.0159. The opinions expressed and arguments employed herein do not necessarily reflect the official views of these funding bodies.

## Footnotes

[1]`https://github.com/BayesWatch/deep-kernel-transfer`

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
