[Supplementary Material]

# A Kernels

**Polynomial.** This computes a covariance matrix based on the Polynomial kernel between inputs

$$k'(x, x') = (x^\top x' + c)^p,$$  (9)

where $p$ is the degree of the polynomial and $c$ is an offset parameter. We used $p = 1$ and $p = 2$ in our experiments.

**Radial Basis Function kernel (RBF).** The RBF is a stationary kernel given by the squared Euclidean distance between the two inputs

$$k'(x, x') = \exp\left(-\frac{||x - x'||^2}{2l^2}\right),$$  (10)

where $l$ is a lengthscale parameters learned at training time.

**Matérn kernel.** This is a stationary kernel which is a generalization of the RBF and the absolute exponential kernel. It is parameterized by a value $\nu > 0$, commonly chosen as $\nu = 1.5$ (giving once-differentiable functions) or $\nu = 2.5$ (giving twice differentiable functions). The kernel is defined as follows:

$$k'(x, x') = |x - x'|^\nu K_\nu(|x - x'|).$$  (11)

We used a value of $\nu = 2.5$ in our experiments.

**Spectral mixture kernel.** The spectral mixture kernel was introduced by Wilson and Adams (2013) as a powerful stationary kernel for estimating periodic functions. The kernel models a spectral density with a Gaussian mixture

$$k'(\tau) = \sum_{q=1}^{Q} w_q \prod_{p=1}^{P} \exp\left\{-2\pi^2 \tau_p^2 v_q^{(p)}\right\} \cos\left(2\pi \tau_p \mu_q^{(p)}\right),$$  (12)

where $\tau = x - x'$, $w_q$ are weights that specify the contribution of each mixture component, $\mu_q$ are the component periods, and $v_q$ are lengthscales determining how quickly a component varies with the inputs $x$. We used 4 mixtures in our experiments.

**Cosine similarity kernel (CosSim).** The cosine similarity kernel consists in taking the product between the unit-normalized input vectors

$$k'(x, x') = \frac{xx'}{||x|| \, ||x'||}.$$  (13)

The cosine similarity ranges from -1 (opposite) to 1 (same), with 0 indicating decorrelation (orthogonal). Following the suggestions in Wang et al. (2019) we experimented with another variant, meaning centering the input vectors through BatchNorm (BN) statistics Ioffe and Szegedy (2015) before the normalization (BNCosSim).

# B Training Details

**Datasets.** The CUB dataset (Wah et al., 2011) consists of 11788 images across 200 classes. We divide the dataset in 100 classes for train, 50 for validation, and 50 for test (Hilliard et al., 2018; Chen et al., 2019). The mini-ImageNet dataset (Ravi and Larochelle, 2017) consists of a subset of 100 classes (600 images for each class) taken from the ImageNet dataset (Russakovsky et al., 2015). We use 64 classes for train, 16 for validation and 20 for test, as is common practice (Ravi and Larochelle, 2017; Chen et al., 2019). The Omniglot dataset (Lake et al., 2011) contains 1623 black and white characters taken from 50 different languages. Following standard practice, the number of classes is increased to 6492 by adding examples rotated by $90°$, and we use 4114 for training. The EMNIST dataset (Cohen et al., 2017) contains single digits and characters from the English alphabet. We split the 62 classes into 31 for validation and 31 for test.

**Regression.** In the function prediction experiment, we use the same backbone network described in Finn et al. (2017): a two-layer MLP, where each layer has 40 units and ReLU activations. We use the Adam optimizer with learning rate $10^{-3}$ over $5 \times 10^5$ training iterations. For regression with feature transfer, a network is trained to predict the output of a function over all tasks, before being fine-tuned on a new task (with 1 or 100 steps of size $10^{-3}$). For the head pose estimation backbone, we use a three-layer convolutional neural network, each with 36 output channels, stride 2, and dilation 2 to downsample the $100 \times 100$ input images. We train for 100 steps using the Adam optimizer with learning rate $10^{-3}$.

**Classification.** At training time we apply standard data augmentation (random crop, horizontal flip, and color jitter). The 1-shot training consists of 600 epochs, and 5-shot of 400, for MAML it corresponds to 60000 and 40000 episodes, and for Feature Transfer and Baseline++ to 400 and 600 supervised epochs with a mini-batch size of 16. In DKT, the hyperparameters of the kernel are optimized with a learning rate one order of magnitude lower than that used for training the CNN. This helped with convergence. In all experiments we used first-order MAML for memory efficiency. This does not significantly affect results (see Chen et al., 2019). In all cases the

validation set has been used to select the training epoch/episode with the best accuracy. In classification and cross-domain experiments, each method uses the same backbone (a four layer CNN), optimizer (Adam), and learning rate ($10^{-3}$). We use shallow backbones because they have been shown to highlight differences between methods (Chen et al., 2019). The CNN used for classification is given in Figure 2.

Figure 2: The CNN used as a backbone for classification. It consists of 4 convolutional layers, each consisting of a 2D convolution, a batch-norm layer, and a ReLU non-linearity. The first convolution changes the the number of channels of the input to 64, and the remaining convolutions retain this channel dimension. Each convolutional layer is followed by a max-pooling operation that decreases the spatial resolution of its input by a half. Finally, the output is flattened into a vector when is used as a feature.

## C    Additional Results: Regression Experiments

Here, we provide additional samples of the few-shot regression experiments for a qualitative comparison (Figure 3). Additionally we compare the latent spaces in the head trajectory estimation experiment. We reduced the number of hidden units to $\mathbf{h} = \{h_1, h_2\}$ and used a hyperbolic tangent activation function (tanh) to project the values to a Cartesian plane with $h_i \in [-1, 1]$. We then sampled 100 trajectories from the test set and recorded the value of $\mathbf{h}$ for the targets. The resulting plot is shown in Figure 4. The spectral kernel enforces a more compact manifold, clustering the head poses on a linear gradient based on the value of the target, leading to more accurate predictions.

Figure 3: Additional samples for the unknown periodic function prediction experiment. We compare methods for in-range (top row) and out-of-range (bottom row) conditions. The true function is plotted in solid blue, the out-of-range portion in dotted blue, the approximation in red, and the uncertainty is given by a red shadow. The 5 support points (blue stars) are uniformly sampled in the available range.

Figure 4: Latent space representation enforced by an RBF (left) and Spectral (right) kernel on the head trajectory experiments.

# D   Additional Results: Classification Experiments

Table 4: Average accuracy and standard deviation (percentage) on the few-shot classification setting (5-ways). [top] Results reported in recent literature. For a fair comparison we selected only those methods that have been trained with a similar backbone and training schedule. [center-bottom] Methods trained from scratch (three runs) with the same backbone (a four layer CNN), optimizer (Adam), and learning rate ($10^{-3}$). Test performed on novel classes with 3000 randomly generated tasks. DKT is competitive across various datasets and conditions. Best results highlighted in bold. *Reported by Jerfel et al. (2019) using a comparable backbone.

| | CUB | | mini-ImageNet | |
|---|---|---|---|---|
| **Method** | **1-shot** | **5-shot** | **1-shot** | **5-shot** |
| **ML-LSTM** (Ravi and Larochelle, 2017) | – | – | $43.44 \pm 0.77$ | $60.60 \pm 0.71$ |
| **SNAIL** (Mishra et al., 2018) | – | – | 45.10 | 55.20 |
| **iMAML-HF** (Rajeswaran et al., 2019) | – | – | $49.30 \pm 1.88$ | – |
| **LLAMA** (Grant et al., 2018) | – | – | $49.40 \pm 1.83$ | – |
| **VERSA** (Gordon et al., 2019)* | – | – | $48.53 \pm 1.84$ | – |
| **Amortized VI** (Gordon et al., 2019) | – | – | $44.13 \pm 1.78$ | $55.68 \pm 0.91$ |
| **Meta-Mixture** (Jerfel et al., 2019) | – | – | $49.60 \pm 1.50$ | $64.60 \pm 0.92$ |
| **SimpleShot** (Wang et al., 2019) | – | – | $49.69 \pm 0.19$ | $\mathbf{66.92 \pm 0.17}$ |
| **Feature Transfer** | $46.19 \pm 0.64$ | $68.40 \pm 0.79$ | $39.51 \pm 0.23$ | $60.51 \pm 0.55$ |
| **Baseline++** (Chen et al., 2019) | $61.75 \pm 0.95$ | $\mathbf{78.51 \pm 0.59}$ | $47.15 \pm 0.49$ | $66.18 \pm 0.18$ |
| **MatchingNet** (Vinyals et al., 2016) | $60.19 \pm 1.02$ | $75.11 \pm 0.35$ | $48.25 \pm 0.65$ | $62.71 \pm 0.44$ |
| **ProtoNet** (Snell et al., 2017) | $52.52 \pm 1.90$ | $75.93 \pm 0.46$ | $44.19 \pm 1.30$ | $64.07 \pm 0.65$ |
| **MAML** (Finn et al., 2017) | $56.11 \pm 0.69$ | $74.84 \pm 0.62$ | $45.39 \pm 0.49$ | $61.58 \pm 0.53$ |
| **RelationNet** (Sung et al., 2018) | $62.52 \pm 0.34$ | $78.22 \pm 0.07$ | $48.76 \pm 0.17$ | $64.20 \pm 0.28$ |
| **DKT + CosSim** (ours) | $\mathbf{63.37 \pm 0.19}$ | $77.73 \pm 0.26$ | $48.64 \pm 0.45$ | $62.85 \pm 0.37$ |
| **DKT + BNCosSim** (ours) | $62.96 \pm 0.62$ | $77.76 \pm 0.62$ | $\mathbf{49.73 \pm 0.07}$ | $64.00 \pm 0.09$ |

**Kernel comparison.** In Table 5 we show a comparison between different kernels (linear, RBF, Matérn, Polynomial $p = 1$ and $p = 2$, CosSim, BNCosSim) trained on CUB and mini-ImageNet. In this setting using a BNCosSim kernel gives a large advantage in almost all conditions. This result is in line with the findings of Wang et al. (2019), who showed how centering and unit normalizing the features considerably improve the performance in classification tasks. The overall performance of CosSim and BNCosSim is also in accordance with the findings of Chen et al. (2019) and their implementation of Baseline++, an effective feature transfer method based on the cosine distance. Further investigations are necessary in this direction to understand the reason why cosine metrics and normalization are so important in few-shot learning.

Table 5: Average accuracy and standard deviation (percentage) on the few-shot classification setting (5-ways) for different kernels. Methods trained from scratch (three runs) with the same backbone (a four layer CNN), optimizer (Adam), and learning rate ($10^{-3}$). Test performed on novel classes with 3000 randomly generated tasks.

| | CUB | | mini-ImageNet | |
|---|---|---|---|---|
| **Kernel** | **1-shot** | **5-shot** | **1-shot** | **5-shot** |
| **Linear** | $60.23 \pm 0.76$ | $74.74 \pm 0.22$ | $48.44 \pm 0.36$ | $62.88 \pm 0.46$ |
| **RBF** | $55.34 \pm 2.56$ | $73.20 \pm 1.41$ | $45.92 \pm 1.08$ | $61.42 \pm 0.74$ |
| **Matérn** | $58.20 \pm 0.63$ | $73.21 \pm 1.30$ | $47.65 \pm 0.85$ | $62.59 \pm 0.12$ |
| **Polynomial** ($p = 1$) | $59.54 \pm 1.10$ | $74.51 \pm 0.98$ | $47.78 \pm 0.60$ | $62.54 \pm 0.96$ |
| **Polynomial** ($p = 2$) | $5718 \pm 0.40$ | $71.14 \pm 0.58$ | $46.36 \pm 0.34$ | $60.26 \pm 0.40$ |
| **CosSim** | $\mathbf{63.37 \pm 0.19}$ | $77.73 \pm 0.26$ | $48.64 \pm 0.45$ | $62.85 \pm 0.37$ |
| **BNCosSim** | $62.96 \pm 0.62$ | $\mathbf{77.76 \pm 0.62}$ | $\mathbf{49.73 \pm 0.07}$ | $\mathbf{64.00 \pm 0.09}$ |

Table 6: Average accuracy and standard deviation (percentage) over three runs on 1-shot and 5-shot classification (5-ways), for different backbones in the CUB dataset. We use the same setup as in the classification setting. The results for the ResNet are the ones reported in Chen et al. (2019). DKT has the best score in 1-shot Conv-4, and 5-shot ResNet, while being competitive in the other conditions. Best results highlighted in bold.

| Method | Conv-4 | | ResNet-10 | |
| --- | --- | --- | --- | --- |
| | 1-shot | 5-shot | 1-shot | 5-shot |
| Feature Transfer | $46.19 \pm 0.64$ | $68.40 \pm 0.79$ | $63.64 \pm 0.91$ | $81.27 \pm 0.57$ |
| Baseline++ (Chen et al., 2019) | $61.75 \pm 0.95$ | $\mathbf{78.51 \pm 0.59}$ | $69.55 \pm 0.89$ | $85.17 \pm 0.50$ |
| MatchingNet (Vinyals et al., 2016) | $60.19 \pm 1.02$ | $75.11 \pm 0.35$ | $71.29 \pm 0.87$ | $83.47 \pm 0.58$ |
| ProtoNet (Snell et al., 2017) | $52.52 \pm 1.90$ | $75.93 \pm 0.46$ | $\mathbf{73.22 \pm 0.92}$ | $85.01 \pm 0.52$ |
| MAML (Finn et al., 2017) | $56.11 \pm 0.69$ | $74.84 \pm 0.62$ | $70.32 \pm 0.99$ | $80.93 \pm 0.71$ |
| RelationNet (Sung et al., 2018) | $62.52 \pm 0.34$ | $78.22 \pm 0.07$ | $70.47 \pm 0.99$ | $83.70 \pm 0.55$ |
| DKT + CosSim (ours) | $\mathbf{63.37 \pm 0.19}$ | $77.73 \pm 0.26$ | $70.81 \pm 0.52$ | $83.26 \pm 0.50$ |
| DKT + BNCosSim (ours) | $62.96 \pm 0.62$ | $77.76 \pm 0.62$ | $72.27 \pm 0.30$ | $\mathbf{85.64 \pm 0.29}$ |

Table 7: Average *Expected Calibration Error* (*ECE*, Guo et al. 2017) with standard deviation (percentage) over three runs on 1-shot and 5-shot classification (5-ways) in the CUB dataset. The lower the better. For the training phase we used the same setup as in the classification experiments. In the evaluation phase, the temperature of all models has been calibrated on 3000 randomly generated tasks, then each method has been evaluated on a separate set of 3000 randomly generated test tasks. DKT has the third lowest error in 1-shot, and the second lowest error in 5-shot. Best results highlighted in bold.

| Method | 1-shot | 5-shot |
| --- | --- | --- |
| Feature Transfer | $12.57 \pm 0.23$ | $18.43 \pm 0.16$ |
| Baseline++ (Chen et al., 2019) | $4.91 \pm 0.81$ | $2.04 \pm 0.67$ |
| MatchingNet (Vinyals et al., 2016) | $3.11 \pm 0.39$ | $2.23 \pm 0.25$ |
| ProtoNet (Snell et al., 2017) | $\mathbf{1.07 \pm 0.15}$ | $\mathbf{0.93 \pm 0.16}$ |
| MAML (Finn et al., 2017) | $1.14 \pm 0.22$ | $2.47 \pm 0.07$ |
| RelationNet (Sung et al., 2018) | $4.13 \pm 1.72$ | $2.80 \pm 0.63$ |
| DKT + BNCosSim (ours) | $2.62 \pm 0.19$ | $1.15 \pm 0.21$ |

# E   Additional Results: Cross-Domain Experiments

Table 8: Average accuracy and standard deviation (percentage) over three runs on the cross-domain setting (5-ways). We use the same setup as in the classification setting. The proposed method (DKT) has the best score on most conditions. Best results highlighted in bold.

| Method | Omniglot→EMNIST | | mini-ImageNet→CUB | |
| --- | --- | --- | --- | --- |
| | 1-shot | 5-shot | 1-shot | 5-shot |
| Feature Transfer | $64.22 \pm 1.24$ | $86.10 \pm 0.84$ | $32.77 \pm 0.35$ | $50.34 \pm 0.27$ |
| Baseline++ (Chen et al., 2019) | $56.84 \pm 0.91$ | $80.01 \pm 0.92$ | $39.19 \pm 0.12$ | $\mathbf{57.31 \pm 0.11}$ |
| MatchingNet (Vinyals et al., 2016) | $75.01 \pm 2.09$ | $87.41 \pm 1.79$ | $36.98 \pm 0.06$ | $50.72 \pm 0.36$ |
| ProtoNet (Snell et al., 2017) | $72.04 \pm 0.82$ | $87.22 \pm 1.01$ | $33.27 \pm 1.09$ | $52.16 \pm 0.17$ |
| MAML (Finn et al., 2017) | $72.68 \pm 1.85$ | $83.54 \pm 1.79$ | $34.01 \pm 1.25$ | $48.83 \pm 0.62$ |
| RelationNet (Sung et al., 2018) | $75.62 \pm 1.00$ | $87.84 \pm 0.27$ | $37.13 \pm 0.20$ | $51.76 \pm 1.48$ |
| DKT + Linear (ours) | $\mathbf{75.97 \pm 0.70}$ | $89.51 \pm 0.44$ | $38.72 \pm 0.42$ | $54.20 \pm 0.37$ |
| DKT + CosSim (ours) | $73.06 \pm 2.36$ | $88.10 \pm 0.78$ | $\mathbf{40.22 \pm 0.54}$ | $55.65 \pm 0.05$ |
| DKT + BNCosSim (ours) | $75.40 \pm 1.10$ | $\mathbf{90.30 \pm 0.49}$ | $40.14 \pm 0.18$ | $56.40 \pm 1.34$ |

**Kernel comparison.** In Table 9 we show a comparison between different kernels (linear, RBF, Matérn, Polynomial $p = 1$ and $p = 2$, CosSim, BNCosSim) trained on Omniglot→EMNIST and mini-ImageNet→CUB. Overall using a BNCosSim kernel still gives an advantage in almost all conditions, showing stable results. The best accuracy is achieved using more specialized kernels, however they often reach peak performance in specific conditions while underperforming in others.

Table 9: Average accuracy and standard deviation (percentage) over three runs on the cross-domain setting (5-ways) for different kernels. We use the same setup as in the classification setting.

| Kernel | Omniglot→EMNIST | | mini-ImageNet→CUB | |
|---|---|---|---|---|
| | 1-shot | 5-shot | 1-shot | 5-shot |
| **Linear** | **75.97 ± 0.70** | 89.51 ± 0.44 | 38.72 ± 0.42 | 54.20 ± 0.37 |
| **RBF** | 74.46 ± 0.41 | 88.38 ± 0.53 | 36.22 ± 0.40 | 51.30 ± 0.52 |
| **Matérn** | 75.46 ± 0.20 | 88.04 ± 1.81 | 36.98 ± 0.41 | 51.35 ± 0.16 |
| **Polynomial** ($p = 1$) | 74.33 ± 0.67 | **90.72 ± 0.47** | 38.24 ± 0.30 | 54.11 ± 0.40 |
| **Polynomial** ($p = 2$) | 75.58 ± 1.18 | 88.06 ± 0.70 | 36.83 ± 0.46 | 51.92 ± 0.87 |
| **CosSim** | 73.06 ± 2.36 | 88.10 ± 0.78 | **40.22 ± 0.54** | 55.65 ± 0.05 |
| **BNCosSim** | 75.40 ± 1.10 | 90.30 ± 0.49 | 40.14 ± 0.18 | **56.40 ± 1.34** |