[Reviews · NeurIPS 2020]

Review 1

Summary and Contributions: This paper investigates augmenting Gaussian process models with meta-learned neural network features, similarly to deep kernels. These features are trained via type 2 maximum likelihood. The authors investigate this approach in the regression setting (standard setting for GPs) and in a one versus all classification setting. The approach is evaluated on standard benchmarks (sinusoid, CUB, miniImagenet) plus a head pose regression task. The authors also investigate cross domain transfer, where the method is trained on one dataset and transfered to another.

Strengths: Overall, the paper is clear and fairly strong. The method is straightforward, but the presentation is clear and the investigation is reasonably thorough. The cross-domain experiments are interesting, as is the approach for classification. The investigation of various kernels is a strength of the work.

Weaknesses: There are a few weaknesses which, if addressed, would strengthen the paper. If addressed, I would increase my score. These are: - Why do the authors not investigate ADKL, R2D2 and ALPaCA on the out-of-range experimental setting? This seems like a minor experiment to run to improve the completeness of the experiments, if these models are already implemented. The same is true for the head pose trajectory experiment. While these may seem like minor considerations, the meta-learning landscape is now busy enough that rigorous comparisons to baselines is necessary for progress in the field. - Since one of the strengths of the method (as claimed by the authors) is uncertainty characterization, the authors should include a quantitative evaluation of the uncertainty. For example, the authors could report log likelihood for methods that output a density, and could evaluate e.g. model calibration.

Correctness: The paper appears to be correct. The evaluation procedure is standard for the few-shot learning literature.

Clarity: The paper is clearly written.

Relation to Prior Work: The paper continues a line of work looking at the intersection of empirical Bayes and meta-learning. In particular, the method relies on Gaussian conjugacy to simplify the inner optimization problem (in an optimization-based meta-learning framework) to an analytically tractable one. While the authors discuss ALPaCA and R2D2, it should be made more clear that the analytically tractable inner optimization problem was presented in these works. In particular, ALPaCA similarly leverages Gaussian conjugacy, just in the parametric Bayesian least squares setting as opposed to the non-parametric setting. Other than this minor issue, the discussion of the prior work is reasonable. -----Post-Rebuttal----- The authors have added a nice set of experiments as requested. I believe these experiments strengthen the paper, so I will increase my score.

Reproducibility: Yes

Additional Feedback: None


Review 2

Summary and Contributions: The paper proposes a simple and well-motivated adaptation of deep kernel learning (DKL) [1] for few-shot meta-learning. The idea is to train a deep kernel that's shared for all tasks, and integrate out the parameters within each task via using Gaussian processes. The authors demonstrate strong empirical results with the proposed method.

Strengths: - I really like the idea of the paper. It's simple, well-motivated, and seems to work well. This is one of the most interesting applications of the deep kernel learning idea that I have seen. - I am not an expert in meta / few-shot learning, but the empirical results appear to be strong. The authors do a thorough comparison with a range of strong baselines and achieve better performance.

Weaknesses: The method likely inherits the issues of DKL. In particular, the inference on classification is not exact, and approximations have to be made. The complexity in terms of the number of data points within each task is cubic, which is not an issue in the few-shot setting, but can be an issue more generally. Approximations have to be made to scale the method to larger tasks. Another issue with DKL can be that a powerful backbone network can overfit even when trained through the GP marginal likelihood: the backbone network can almost solve the tasks and provide embeddings that lead to very simple GP fits; this can in turn lead to poor uncertainty calibration. None of these issues seem to affect the performance in the experiments in the paper, but could theoretically hinder performance more generally.

Correctness: To the best of my understanding the presented method is sound, and the experiments are conducted correctly.

Clarity: The paper is written very clearly, and easy to follow.

Relation to Prior Work: The paper does a good job explaining the relation to prior work.

Reproducibility: Yes

Additional Feedback: Question: - Did you encounter any issues with the backbone network overfitting in your experiments, especially on harder problems with images? - Did you use any regularization to prevent overfitting with the backbone network? ---------------------------------------------- Post-rebuttal I have read the other reviews and the feedback from the authors and I would like to maintain my assessment. I think this is a good paper and I recommend acceptance. [1] Wilson, Andrew Gordon, et al. "Deep kernel learning." Artificial intelligence and statistics. 2016.


Review 3

Summary and Contributions: This paper proposes a novel meta-learning approach for the few-shot setting. The key idea is it learns parameters and some hyperparameters of a task-common network while marginalizing over the task-specific parameters. The technique is easy to implement and is capable of quantifying uncertainty.

Strengths: * proposes an efficient and theoretically sound few-shot learning technique that takes into account uncertainty * satisfactory amount of experimental evaluation * discusses how it relates to several other meta-learning techniques e.g. the connection between marginalizing over the task-specific parameters and the inner-loop in MAML.

Weaknesses: * The rationale for certain choices is not clear. * The uncertainty is not assessed.

Correctness: The motivation is sound and theoretical derivations are correct.

Clarity: Clear enough.

Relation to Prior Work: Good but lacks discussion on other uncertainty-aware meta-learning techniques. See additional feedback.

Reproducibility: Yes

Additional Feedback: I enjoyed reading this paper. * The paper variously highlights deep kernel learning, even in the title. Is using deep kernel learning a contribution of the paper? I believe the proposed method is applicable for any Gaussian process-style models. Where deep kernels help is to work with high-dimensional inputs such as images. Vanilla Gaussian processes (GP) are more suitable for a few data points due to the cubic computational complexity whereas deep networks are more suitable for big-data settings. This means, to train a deep kernel, we need a lot of data in the front-end which is not preferred by the GP. * It is not clear if the uncertainty quantification is indeed accurate as it has not been evaluated, though visually seems to be acceptable in the plots shown. Clarify why the proposed approach works better than ALPaCA. * I’d like authors to discuss other uncertainty-aware transfer learning techniques based on neural processes and optimal transport. - MetaFun: Meta-Learning with Iterative Functional Updates, arXiv, 2020 - Online Domain Adaptation for Occupancy Mapping, RSS, 2020 I do understand that these papers were “officially published” after the NeurIPS submission deadline. Therefore, a discussion is sufficient, though experimental comparisons are more than welcome. * Define “Small but related tasks.” - how to measure the relatedness? * Define “dispersive” * “Needs to estimate higher-order derivatives” - it is not clear why. *Move algorithm 1 to the main paper from the appendix as it helps to grasp the high-level idea of the algorithm quickly. ### POST-REBUTTAL COMMENTS ### Thanks for the rebuttal! The rebuttal would have been much stronger if the limited space was dedicated to explicitly answer all questions rather than unnecessarily highlighting and reiterating the positive aspects of reviews. As R1 and myself have highlighted, I still feel that it is required to evaluate the uncertainty. I'd like to keep my score unchanged.


Review 4

Summary and Contributions: The authors provided a Bayesian treatment for meta-learning inner loop by using deep kernels. Existing work indicate that given a new unseen task it is possible to estimate the posterior distribution over a query set conditioned on a small support set. In a meta-learning framework it corresponds to a Bayesian treatment for the inner loop cycle. Inspired by these theories, the authors proposed a new approach refer to as deep kernel learning with transfer, or Deep Kernel Transfer (DKT) for short. In addition, the authors derived two classes of DKT for the regression and classification and conduct the compared experiments on regression and classification.

Strengths: The authors provided a new method to treat the few-shot learning scenarios. In meta-learning as hierarchical model, they learned a set of parameters and hyperparameters of a deep kernel (outer-loop) that maximize a marginal likelihood across all tasks. This method is simple because the inner loop can be replaced by a marginal likelihood computation and the method can be implemented as a single optimizer. And it is also efficient in the low-data regime. Experimental results shows the flexibility for different settings and the robustness of this method.

Weaknesses: This idea is too simple and easy to think of. This proposal just replaces the inner loop by a marginal likelihood computation while still optimize the parameters in meta-learning as hierarchical model. For biased data this method may not provide a correct estimation.

Correctness: Yes, in my opinion, the claims and method are all correct. And the empirical methodology is also correct.

Clarity: I can understand what the authors tried to say but some of the expressions are not very idiomatic.

Relation to Prior Work: Yes, the authors discussed clearly about the difference between this method and previous work

Reproducibility: Yes

Additional Feedback: Post_rebuttal I have read the rebuttal from authers, the authors replied to my question clearly. I will increase my overall score.

[Author Response · NeurIPS 2020]

We thank the reviewers for their detailed comments on our manuscript. We are glad that the response has been positive.
Below we respond to each reviewer individually, providing additional considerations (and data).

**Reviewer #1** recommends acceptance, highlighting the clarity of the paper, the interesting approach to classification,
and useful investigations of the cross-domain setting and kernel analysis. The reviewer pointed out two weaknesses,
stating that if they are addressed, this would increase the reviewer's score. We thank the reviewer for the positive
feedback and suggestions for improving the work; we have addressed the requests as best as we could given the short
deadline. *(1) Experiments for out-of-range condition in ADKL, R2D2 and ALPaCA.* The code of ADKL is not available
and the code of R2D2 is only available for the classification case. For the in-range condition we have reported the
results from Tossou et al. (2019, reference in the paper). For ALPaCA there is public code for the in-range condition
(used to generate the MSE in Table 1). Following the request of the reviewer we have now modified the ALPaCA
code for out-of-range; the score of ALPaCA is $5.92 \pm 0.11$ (MSE, average of three runs). ALPaCA is better than
MAML ($8.45$ MSE) but significantly less accurate than our method (DKL has a lower MSE, $0.14$ and $0.11$). We
will add the new results to Table 1 and mark with an asterisk that the ADKL-R2D2 scores are from Tossou et al.
(2019). Porting the code of ALPaCA for the head-pose regression has been technically challenging given the short
deadline (it is based on completely different libraries). *(2) Quantitative evaluation of the uncertainty.* Following the
suggestion of the reviewer we ran experiments on model calibration. We have followed the protocol of Guo et al. (2017,
"On Calibration of Modern Neural Networks") which consists in temperature scaling plus estimation of the Expected
Calibration Error (ECE), a scalar summary statistic (the lower the better). For each method, we found the temperature
by minimizing NLL on logits/labels via LBFGS on 3000 tasks; then we estimated the ECE on the test set. We report the
results on the CUB dataset in the format *"1-shot/5-shot"* (percentage, average of three runs): DKL+BNCosSim (ours)
$2.6/1.1$, Feature Transfer $12.6/18.4$, Baseline++ $4.9/2.0$, MatchingNet $3.1/2.2$, ProtoNet $1.1/0.9$, MAML $1.1/2.5$,
RelationNet $4.1/2.8$. In 1-shot our model achieves one of the lowest ECE $2.6\%$ beating most of the competitors (only
ProtoNet and MAML do better). In 5-shot our model achieves the second lowest ECE $1.1\%$ (ProtoNet does marginally
better). These results provide additional evidence about the strength of the method in uncertainty estimation. We will
include tables in the appendix and a short summary in the paper.

**Reviewer #2** marked the paper for a clear acceptance, highlighting that the method is well motivated and performs well
in experimental comparisons. We thank the reviewer for the positive feedback which has captured the essential strength
of this work. Here we provide compact answers to the two questions. *(1) Potential overfitting.* For the standard conv-4
used in few-shot learning we did not encounter any overfitting problems. We have also tried a ResNet-10 (see Table 6,
supp. material) and similarly we did not observe overfitting. *(2) Use of regularization.* Following common practice in
few-shot literature, we have used data augmentation as the only form of regularization (for all methods). Note that, a
strength of our Bayesian approach is that it applies ML-II which has an implicit regularization effect.

**Reviewer #3** marked the paper as a good submission and recommended acceptance, highlighting as strengths the
efficiency of the method, the theoretical grounding, and the empirical evaluation. We thank the reviewer for the
rich commentary and considerations. *(1) Possibility of applying the method to Gaussian process-style models.* The
reviewer is right, the method could be easily extended to a standard Gaussian process by finding the parameters of the
kernel. *(2) On why the proposed approach works better than ALPaCA.* Our analysis suggests that ALPaCA may suffer
the shortcomings of complex meta-learning methods, being harder to train and less flexible in domain transfer and
out-of-distribution fitting. An additional experiment seems to confirm this hypothesis (see answer to Reviewer #1). In
the out-of-range condition ALPaCA has a higher MSE compared to our method ($5.92$ vs $0.11$). Qualitative analyses
showed that ALPaCA is not able to provide a good fit for points out of its training range, with a behavior similar to
MAML (see Figure 1a in the paper). *(3) Other methods.* The reviewer asked for discussion of two uncertainty-aware
transfer learning methods. We thank the reviewer for indicating those papers, we will discuss them in the "Related Work"
section. *(4) Clarifications.* (i) "Small but related tasks", tasks sampled from a common distribution. (ii) "Dispersive",
colliding terminologies used in few-shot literature. (iii) "Needs to estimate higher-order derivatives", in MAML it is
necessary to estimate the gradient of the gradient in the training loop. *(5) Move Algorithm 1 to the main paper.* We
agree; this was our initial intention but given the space constraints we have been unable to do it gracefully. Given this
emphasis by the reviewer, we will try again for the camera ready.

**Reviewer #4** recommends acceptance, pointing out the efficiency and robustness of the method. At the same time the
reviewer identified the simplicity as one of its weaknesses. We discussed this point in the introduction of the paper,
stating how we consider simplicity as a strength. Previous methods are based on complex meta-learning routines, which
we showed can be greatly simplified by adopting our approach (with better performances). The few-shot learning
literature has been saturated with overly complicated solutions, in stark contrast we show that a well designed Bayesian
approach can prune unnecessary meta-learning loops while being very effective in practice. Simple is robust, easy
to implement, debug, reproduce, and apply. Though simple, this method is currently not common practice, and we
believe this paper strongly and rightly emphasizes the need to start with simple approaches before diving in to complex
meta-learning routines. We hope these considerations will resonate with the reviewer.

[Meta-Review · NeurIPS 2020]

The paper provides a nice adaptation of deep kernel learning to the few-shot setting, with promising performance over key deep learning baselines. Reviewers are united in their support for the work. Please carefully consider reviewer comments (and post rebuttal updates) in preparing final revisions.